# One year of attenuation data from a commercial dual-polarized duplex microwave link with concurrent disdrometer, rain gauge and weather observations

Anna Špačková[1], Vojtěch Bareš[1], Martin Fencl[1], Marc Schleiss[3], Joël Jaffrain[4], Alexis Berne[4], and Jörg Rieckermann[2]

[1]Dept. of Hydraulics and Hydrology, Czech Technical University in Prague, Prague, the Czech Republic
[2]Dept. of Urban Water Management, Eawag: Swiss Federal Institute of Aquatic Science and Technology, Dübendorf, Switzerland
[3]Dept. of Geoscience and Remote Sensing, Delft University of Technology, Delft, Netherlands
[4]Environmental Remote Sensing Laboratory, EPFL, Lausanne, Switzerland

**Correspondence:** Anna Špačková (anna.spackova@cvut.cz)

**Abstract.** Commercial microwave links (CML) in telecommunication networks can provide relevant information for remote sensing of precipitation and other environmental variables, such as path-averaged drop size distribution, evaporation, or humidity. The CoMMon field experiment (COmmercial Microwave links for urban rainfall MONitoring) mainly focused on the rainfall observations by monitoring a 38-GHz dual-polarized CML of 1.85 km path length at a high temporal resolution (4 s), as

well as a collocated array of five disdrometers and three rain gauges over one year. The dataset is complemented with observations from five nearby weather stations. Raw and pre-processed data, which can be explored with a custom static HTML viewer, are available at https://doi.org/10.5281/zenodo.4923125 (Špačková et al., 2021). The data quality is generally satisfactory to the further analysis and potentially problematic measurements are flagged to help the analyst identify relevant periods for specific study purposes. Finally, we encourage potential applications and discuss open issues regarding future remote sensing with

CMLs.

## 1   Introduction

Accurate information on precipitation is important for many applications from agriculture to pluvial flooding (Chwala and Kunstmann, 2019). Commercial microwave links (CMLs) from telecommunication networks represent a promising source of information (Graf et al., 2020; Overeem et al., 2016) as their signals are affected by both liquid and solid precipitation.

Presently, there are an estimated 5 million CMLs (Ericsson, 2019) deployed around the world and the widespread coverage of mobile phone networks includes sparsely or completely ungauged areas. CMLs can observe precipitation close to the ground and can be queried remotely from network operation centres within a few seconds enabling operational applications such as precipitation nowcasting or hydrological forecasts and early flood warnings. Currently, only a small number of open datasets for studying the small-scale variability of rainfall and its relationship to CML attenuation data are available. In the Nehterlands,

Van Leth et al. (2018) provided measurements from three collocated microwave links near Wageningen. Gires et al. (2018)

published a dataset of two months of disdrometer data using three collocated devices and provided a dataset of disdrometer data collected during a measurement campaign testing a rainfall simulator (Gires et al., 2020). Unfortunately, datasets of CML attenuation and observations of precipitation microphysics with a high temporal resolution lack the ability to fully tap into this potential and identify current knowledge gaps (see below).

For example, the analysis of Chwala and Kunstmann (2019) suggests that operational datasets from real-world case studies and methods, e.g., for baseline removal, are not openly available. As telecommunication providers are reluctant to share the properties of CMLs, the majority of datasets and important meta information is available only to the research groups involved. However, openly available datasets of precipitation microphysics and CML attenuation, ideally with detailed weather information, are essential to test available theories and benchmark the prediction capabilities of developed methods, which were

mostly tuned and tested on non-public datasets. Nevertheless, there are openly available tools for CML rainfall information processing and mapping (e.g., Overeem et al., 2016; Chwala et al., 2020).

    The goal of this paper is to publish the unique dataset and documentation of the "CoMMon" (Commercial Microwave links for rainfall MONitoring) experiment consisting of a 1-year-long field campaign in Dübendorf (CH), during which attenuation data from a 38-GHz dual-polarized CML were collected, together with precipitation observations from rain gauges and dis-

drometers deployed along the CML path in high temporal resolution (Fig. 1a). In addition, weather data, such as temperature, dew point, relative humidity, and wind speed from two nearby and three more distant weather stations, were acquired. Noteworthy features of the data are: i) a dual-polarized CML, rather than single-polarization; ii) an array of disdrometers in addition to rain gauges; and iii) outdoor units of the CML operated with weather-protecting shields (Fig. 2), to investigate the impact of antenna wetting, for approximately half of the experimental period.

This paper will, first, briefly describe the theory of rain retrieval and highlight the importance of drop size distribution. Second, we present the experimental set up, sensor specifications, experimental campaign, structure of the collected datafiles, and discuss data quality and reliability. The third section presents the database and the html viewer provided to explore the data efficiently. The fourth section discusses potential future applications of the CoMMon dataset.

## 2   Data and methods

**2.1   The importance of drop size distribution for rain retrieval from commercial microwave links**

The attenuation of a CML signal is related to the drop size distribution along its path. The observed attenuation can be used to calculate the rain rate between the two end nodes of a CML. The observed total loss $L_t$ (dB) is the difference of transmitted and received signal power. Rainfall-induced specific attenuation $k$ (dB km$^{-1}$), due to raindrops passing the path of the microwave propagation, can be formulated as:

$k = max(\dfrac{L_t - A_b - A_w}{l}, 0)$                        (1)

where $k$ is the specific attenuation (in dB km$^{-1}$), $A_b$ (dB) is background (baseline) attenuation (ITU-R, 2019), $A_w$ (dB) is wet antenna attenuation and $l$ (km) is path length. $A_b$ is usually determined during dry weather periods (Fencl et al., 2017; Polz et

al., 2020) without dew or rain occurrence (cf. Fig. 7, Fig. 8) and is assumed to have the same level during rainfall. $A_w$ describes the impact of radome wetting. The importance of accurate estimation of $A_w$ increases in the case of short CMLs when its contribution to the observed attenuation is substantial (Pastorek et al., 2018).

The power-law relationship approximates the relation between attenuation caused by raindrops and rainfall intensity (Atlas and Ulbrich, 1977):

$$k = a \cdot R^{\mathrm{b}} \tag{2}$$

where $R$ is the rain rate in mm h$^{-1}$ and parameters $a$ and $b$ are related to the CML characteristics (frequency, polarization) and drop size distribution (DSD) (Olsen et al., 1978). Value $b$ is close to one for frequencies between 20 GHz and 40 GHz (ITU-R, 2005). While electromagnetic scattering for hydrometeors is generally complex (Eriksson, 2018), the specific attenuation for liquid precipitation can be estimated from the drop size distribution:

$$k(f) = 4.343 \times 10^3 \int_D C_{\mathrm{ext}}(D, f) N(D) dD \tag{3}$$

where $f$ is the frequency, $D$ (mm) denotes equi-volumetric drop diameter, $N(D)$ (m$^{-3}$ mm$^{-1}$) is the number of drops per unit volume in a drop diameter interval (drop size distribution, DSD) and $C_{ext}(D, f)$ is the extinction cross section at frequency $f$ in m$^2$ which determines the attenuation from a single drop.

The accuracy of the power-law approximation (Eq. 2) can be assessed by comparing Eq. 3 to the rain rate $R$ (Eq. 4) of the observed drop spectrum (Atlas and Ulbrich, 1977):

$$R = 0.6 \times 10^{-3} \pi \int_D v(D) D^3 N(D) dD \tag{4}$$

where $v(D)$ is the terminal fall velocity (in m s$^{-1}$) of a drop and $D$ (mm) denotes equi-volumetric drop diameter.

## 2.2 Field campaign

The campaign took place in Dübendorf, Switzerland. Figure 1a presents the layout of the CoMMon field campaign with all sites (white pins) where the disdrometers and rain gauges were deployed. The two antennas were located at sites 1 (Dübendorf) and 5 (Wangen) and the CML was 1.85 km long (blue line). To assess wet antenna attenuation, antenna radomes and outdoor units were weather-protected by large custom-made PVC shields for approximately half of the experimental period (Fig. 2). The area between the antennas consists mainly of an airport, sport fields, agricultural fields, a shopping mall and a highway. Five laser optical disdrometers were placed at sites 2, 3, 4 and 5. The disdrometers at site 2 were collocated to enable quality control and the quantification of observation uncertainties. Three tipping bucket rain gauges were also used at sites 2, 4 and 5 (Table 1). The location of each site was chosen based on a compromise between proximity to the CML path, accessibility, and protection against vandalism. The devices were situated as far as possible from roof edges to avoid wind disturbances. The instrument types and specifications are provided in section 2.3 below and in the metadata sheets in the Zenodo repository.

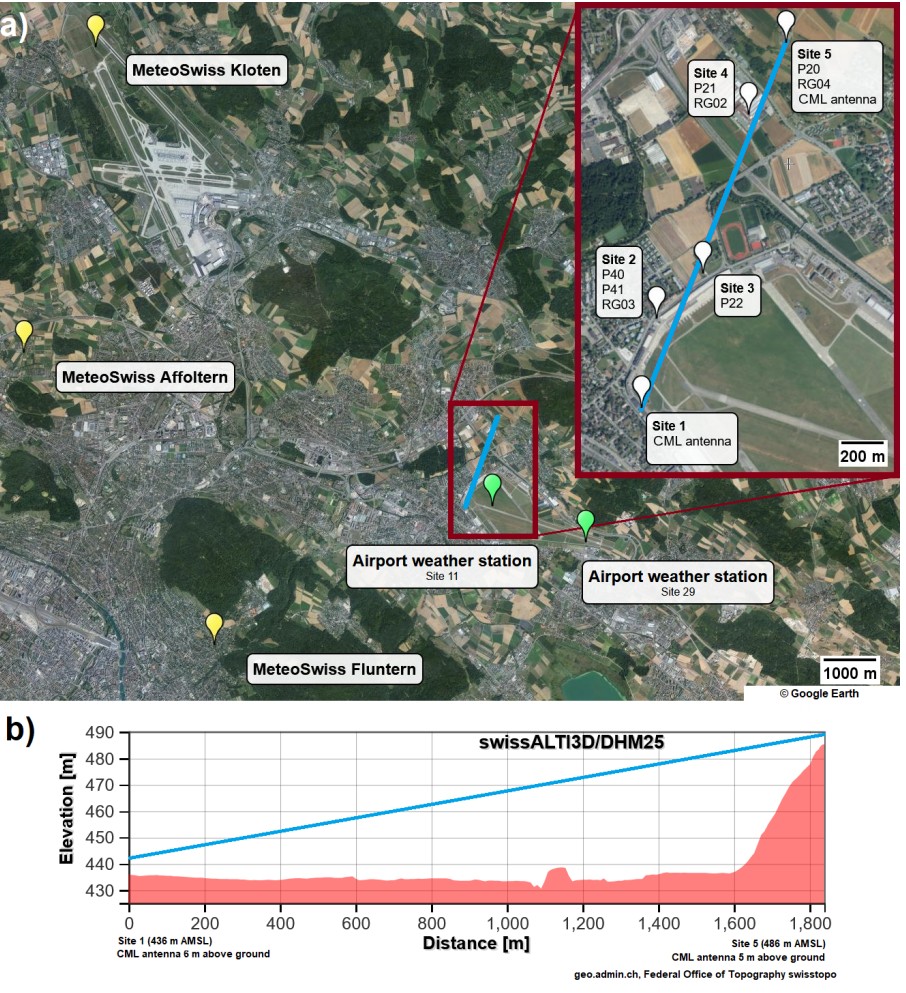

**Figure 1.** a) Layout of the CoMMon field campaign. The radio units are deployed at the end sites (site 1 and site 5). The path of the CML is in blue. Disdrometer (P - Parsivel) and rain gauge (RG - rain gauge) positions are indicated by white pins. Site 2 contained two collocated disdrometers for data quality control. Except for site 3, all rain gauges have a collocated disdrometer. The MeteoSwiss weather stations (yellow pins) are within 6 to 11 km away. Two additional weather stations (green) are located at the Dübendorf airport (site 11 and site 29). b) Longitudinal profile of terrain under the CML path.

The campaign was launched on 9 March 2011 when the disdrometers and tipping bucket rain gauges were deployed. On 17 March 2011 two radio units of the CML were installed. The collected dataset is enhanced by the meteorological data (MeteoSwiss, 2020) obtained from MeteoSwiss (the Federal Office for Meteorology and Climatology in Switzerland) for the duration of the campaign. The weather stations are located in Zürich within 6 to 10 km of the experimental area (Table 2). Moreover, observations from two other weather stations located at the airport (Table 3) provided additional data for the time period from 1 March 2011 to 26 September 2011. The campaign ended on 21 March 2012.

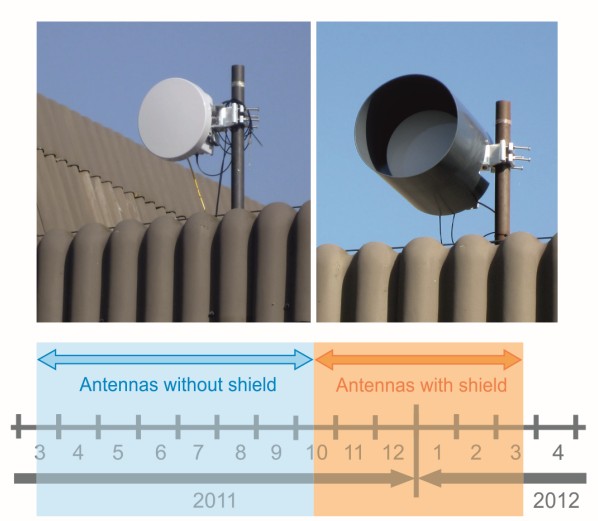

**Figure 2.** (top) Outdoor unit of the Dual-Pol MINI-LINK and the antenna radome without (left) and with custom shield (right). (bottom) Timeline of the operational period of the shielding of antenna radomes.

## 2.3 Instrumentation

### 2.3.1 Commercial microwave link

The CML consisted of two MINI-LINK outdoor units manufactured by Ericsson comprising a radio and a MINI-LINK antenna (ANT20.3 38 HPX, Mod. No. UKY 210 75/DC15 SH) with a diameter of 30 cm. The MINI-LINK is a duplex dual-polarized link which has two communication channels with a horizontal and a vertical polarization. The resolution of the transmitted power (Tx) was 1 dBm and for received power (Rx) it was 1/3 dBm expressed to one decimal place. The horizontally polarized electromagnetic (EM) waves were transmitted at frequencies of 38'657.5 MHz from site 1 and 37'397.5 MHz from site 5. The

vertically polarized EM waves were transmitted at frequencies of 38'650.5 MHz from site 1 and 37'390.5 MHz from site 5. The length of the link was 1850 m. The antenna at site 1 was deployed 6 m above ground and the antenna at site 5 was deployed 5 m above ground. The CML path was not horizontal, the antennas at sites 1 and 5 were located 442 m AMSL and 491 m AMSL respectively (with 1.52° to the horizontal plane). Measurement intervals were changed from the original 15-min-setup to 4 s (instantaneous reading). Data were acquired by a software application based on Simple Network Management Protocol

(Wang et al., 2012). During the measurement campaign, plastic shields were installed on the antennas on 6 October 2011 to avoid water films on the antenna radomes and to eliminate wet antenna attenuation due to rainfall. However, the wet antenna attenuation due to dew cannot be prevented.

     The metadata sheets for the CML can be found in the Zenodo repository.

**Table 1.** Locations of the sites and deployed devices.

| Site | Geographic coordinates | Devices and ID |
|---|---|---|
| 1 | 47°24'4.800" N, 8°37'43.100" E 436 m AMSL | CML antenna |
| 2 | 47°24'17.640"N, 8°37'47.640"E 446 m AMSL | PARSIVEL disdrometers ID: P40 and P41 Tipping bucket rain gauge ID: RG03 |
| 3 | 47°24'24.480"N, 8°37'57.720"E 455 m AMSL | PARSIVEL disdrometer* ID: P22 |
| 4 | 47°24'49.680"N, 8°38'8.520"E 435 m AMSL | PARSIVEL disdrometer ID: P21 Tipping bucket rain gauge ID: RG02 |
| 5 | 47°24'59.760"N, 8°38'16.800"E 486 m AMSL | PARSIVEL disdrometer ID: P20 Tipping bucket rain gauge ID: RG04 CML antenna |

*Note: the only site with unrestricted access

### 2.3.2 Laser optical disdrometers

Raindrop information was collected by the 1$^{st}$ generation of the PARSIVEL optical disdrometer manufactured by OTT Hydromet and retrofitted by EPFL-LTE to allow for remote access and data transfer (Jaffrain et al., 2011). The horizontal laser beam had an area of 54 cm$^2$. The beam was oriented perpendicularly to the dominant wind direction in order to limit undersampling and splashing. The devices were deployed as far as possible from possible sources of turbulence and wake (roof edges and other obstacles). The cob webs were not specifically prevented and they were not observed during the field visits. The mea-

surement principle is based on the attenuation in received voltage and on the time required for the passage of a particle through the laser beam. From this, the terminal fall velocity and the equi-volumetric drop diameter can be estimated. The PARSIVEL rain rate (parameter 05 in Appendix C) retrieval is linked to the drop diameter. Drops larger than 1 mm are assumed to have

**Table 2.** Location of the MeteoSwiss weather stations.

| Weather station | Name | MeteoSwiss STN | Geographic coordinates [m] |
|---|---|---|---|
| ZH_Aff | Zürich Affoltern | 83 | 47°25'39.780"N, 8°31'03.060"E 443 m AMSL |
| ZH_Flun | Zürich Fluntern | 71 | 47°22'41.310"N, 8°33'57.030"E 556 m AMSL |
| ZH_Klo | Zürich Kloten | 59 | 47°28'46.640"N, 8°32'09.890"E 436 m AMSL |

**Table 3.** Location of the airport weather stations.

| Weather station | Geographic coordinates [m] |
|---|---|
| Site 11 | 47°24'06.1194"N, 8°38'09.4164"E |
| Site 29 | 47°23'43.6158"N 8°39'34.2066"E |

an oblate shape with linearly decreasing axis ratio (i.e., the axis ratio between the vertical and horizontal dimensions of the drops) down to a value of 0.7 for 5 mm drops (Battaglia et al., 2010; Löffler-Mang and Joss, 2000). Data were categorized into 115 32 non-equidistant velocity classes and 32 non-equidistant diameter classes (see Appendix A and B). The first two diameter classes were always empty since they were outside the measurement range of the device. The sampling resolution was 30 s.

The metadata sheets for the laser optical disdrometers can be found in the Zenodo repository.

### 2.3.3 Tipping bucket rain gauges

The collocated tipping bucket rain gauges (3029-1, Précis Mécanique) were the same type of model and had been dynamically 120 calibrated (Humphrey et al., 1997). The devices were deployed as far as possible from possible sources of turbulence and wake (roof edges and other structures). Deployed 50 cm from the ground (35 cm of the device itself and 15 cm of piece of wood and concrete), they had a sampling area of 400 cm$^2$. Their bucket content of 4 g corresponded to the resolution of 0.1 mm of rain. The rain gauges were equipped neither with wind protection shields nor with heating, but only 6 snow and 4 mixed precipitation events were observed during the campaign. The loggers had a time resolution 0.1 s and their time drift was less 125 than 2 min per month. The data were saved in the internal memory and downloaded on-site.

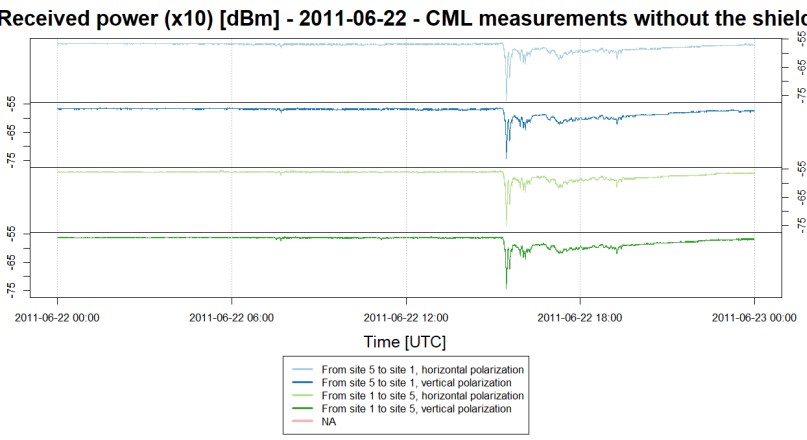

**Figure 3.** Example of measured received power for 22 June 2011 when antennas were not shielded. Green colours label the direction from site 1 to site 5 (D->W) and blue W->D. Light colours depict horizontal polarizations and dark colours vertical polarization. Red vertical lines indicate NA values (not present during this event).

The metadata sheets for the tipping bucket rain gauges can be found in the Zenodo repository.

### 2.3.4 Weather stations

Comprehensive weather data are provided by three additional weather stations operated by MeteoSwiss and located 8.9 km (Affoltern), 5.9 km (Fluntern), and 10.9 km (Kloten) from the experimental site 3 (Fig. 1a) in azimuths 285°, 238°, and 318°, respectively. Data were extracted using CLIMAP software provided by MeteoSwiss. The metadata sheets for the MeteoSwiss stations can be found in the Zenodo repository.

In addition, there were two stations of the Automatic Weather Observation Systems (MIDAS IV, Vaisala) at the Dübendorf airport. The MIDAS IV system employed two sensors at both ends of the runway as these weather stations provide data primarily used for airport operations. The temporal resolution varied between 3 and 60 s depending on the weather parameter. Complete metadata sheets could unfortunately not be compiled from an information provided by Dübendorf airport staff.

### 2.4 Measured variables

CML attenuation data are available between 18 March 2011 and 15 April 2012. The data are given "as is", without any processing or filtering. The columns of the microwave link datafiles are organised as described in Appendix E. The sampling interval of the instantaneous readings was 4 s. Missing observations are denoted by "NA". Figure 3 presents an example of the observed received power for 22 June 2011.

The disdrometers collected data from 11 March 2011 (for P41 from 16 April 2011) to 29 April 2012, and the disdrometer files provided raw data collected by PARSIVELs in 30 s resolutions (see Appendix C for details). The eight types of precipitation (parameter 07 and 08 correspond to precipitation codes according to SYNOP, see Appendix D) were classified based on a

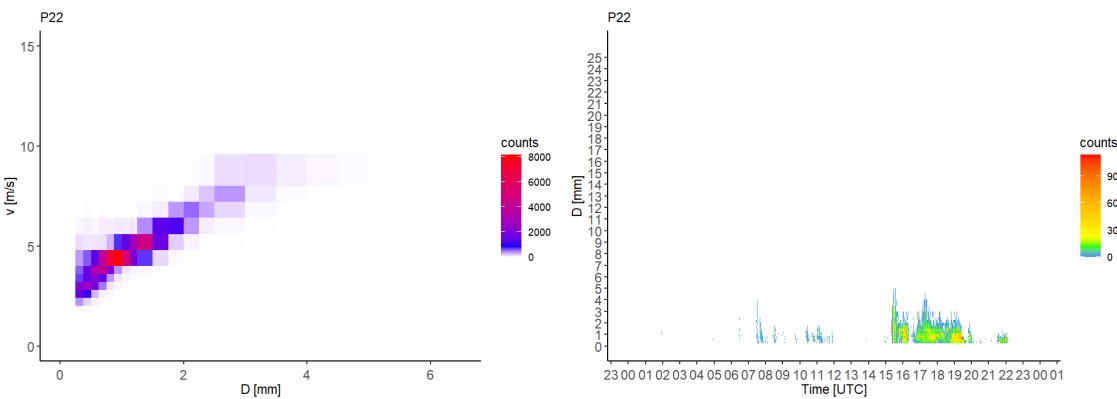

**Figure 4.** Example of disdrometer data for 22 June 2011 at site 3 (disdrometer P22). (left) Scatterplot of the number of drops according to the velocity (terminal fall speed) and size (equi-volumetric drop diameter) classes. (right) Temporal evolution of the DSD.

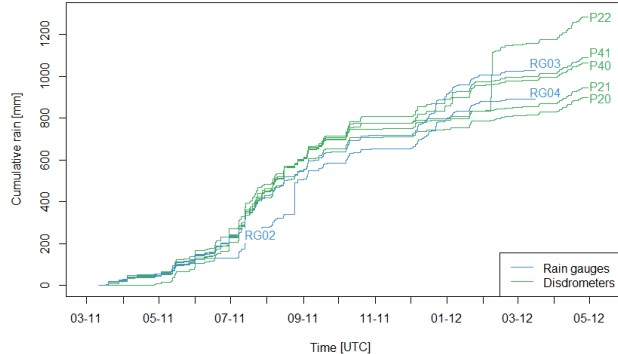

**Figure 5.** Comparison of cumulative rain of the three rain gauges and five disdrometers. There were recorded several outages, suspicious measurements and technical issues, such periods are further described in the Data quality and reliability chapter. Despite this, both types of devices are, in general, in good agreement.

velocity-diameter relationship (Löffler-Mang and Joss, 2000). The code corresponds to the specific precipitation type and its intensity (OTT, 2006). Parameter 16 registered the status of the optical lenses. The raw spectrum of 32x32 drop counts is in parameter 21 (diameter classes x velocity classes). Figure 4 illustrates the observations of the disdrometers.

The tipping bucket rain gauges collected data from 12 March 2011 (for RG04 from 17 March 2011) to 21 March 2012 (for RG04 to 20 March 2012). The rain gauge data had been partly processed. The regular temporal sampling resolution was set to 1 min and the leap year (29 February 2012) had been accounted for. The columns of the datafiles were organised as described in Appendix F. Missing measurements are denoted by "NA".

The comparison of cumulative rain of the three rain gauges and five disdrometers for the whole period of the campaign is presented in Fig. 5. It can be seen that both types of devices are in general in good agreement.

The atmospheric variables of the MeteoSwiss weather stations, collected for the time period from 1 March 2011 to 22 April 2012, are presented in Appendix G. There are 14 parameters in total, including air temperature, air pressure, wind, precipitation, sun radiation, sunlight duration, and dew point. The variables were recorded with a 10-min resolution.

The airport weather station data, collected for the time period from 1 March 2011 to 26 September 2011, contain monitored standard meteorological parameters such as temperature, dew point, relative humidity, pressure, rain intensity, wind speed and direction with columns organised as described in Appendix H.

All measured variables are recorded in UTC time.

## 2.5  Data quality and reliability

Field visits were conducted on 15April 2011, 6 July 2011, 19 October 2011, 1 November 2011 and 16 March 2012 to maintain the instruments. In general, very little maintenance was necessary on the instruments during the frequent field visits. On 15 April 2011, the minor levelling corrections and data collections for all the rain gauges were done. On 6 July 2011, the batteries were replaced in all the rain-gauge data loggers and the data were collected. On 19 October 2011, RG04 was inspected and no problems were found. On 1 November 2011, RG02 and RG04 were inspected with no reported issues. On 16 March 2012, RG02 was found blocked by a leaf and with empty batteries. The RG02 lagged laptop time by approximately 7 minutes. The difference between RG03 time and UTC was -00:05:31. The difference for RG04 was -00:02:28. Only the leap year had been corrected. The rain gauges were dynamically calibrated in the laboratory.

The CML had almost no technical issues causing problems with data collection. Nevertheless, the unit at site 1 (Dübendorf) was not working properly in the period between 16 September 2011 and 9 October 2011 where no data is available.

The disdrometers P20, P21 and P22 did not provide uninterrupted data between 14 October 2011 and 4 November 2011. All disdrometers partially malfunctioned between 7 December 2011 and 14 January 2012; and between 26 January 2012 and 3 February 2012 from the undersupplied solar panels during winter. A significant overestimation of rainfall occurred on 7 and 8 February 2012 when P22 measured around 300 mm of cumulative rainfall. Unfortunately, site 3 did not provide data from a collocated rain gauge for comparison. The filter presented in Jaffrain and Berne (2011) can be used to remove dubious measurements. Disdrometer locations and orientations were chosen to minimize wind disturbances, staying as far away from sharp corners and tall objects as possible. Cob webs were not observed during the field visits.

The rain gauges were deployed in adequate distance from trees and bushes to minimize the influence of wind whirling and clogging by leaves. However, one of the tipping bucket rain gauges faced technical issues that constrained the data collection. The greatest data availability gap, due to low batteries, happened at site 4 (RG02) between 6 July 2011 and 16 March 2012.

Figure 5 presents a comparison of cumulative rain collected by the rain gauges and disdrometers and shows a good temporal match of measured data. The outages of disdrometers between December and February, described above, caused the underestimation of rain amounts during this period. RG02 corresponds to RG03 for the entire time period when it was in operation. RG04 was blocked from the middle of June 2011 until the 6 July 2011. There was also unrealistic rainfall recorded by RG04 on 24 August 2011. In total, 139.7 mm of rainfall were measured within 2.5 h, probably an artefact due to vandalism. The

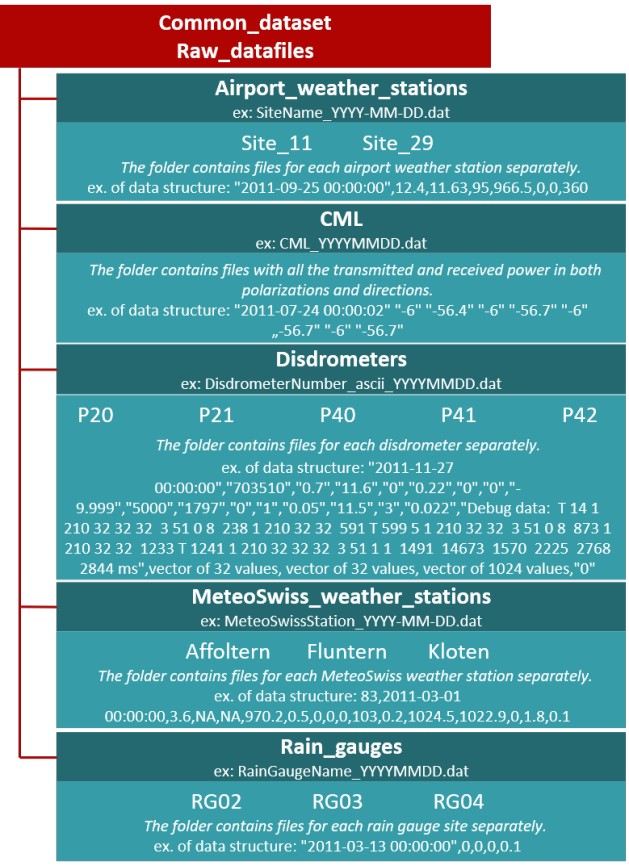

**Figure 6.** The organisation scheme of the files in folders.

collocated disdrometer (P20) also showed a dissimilar temporal evolution of rain rate to the other disdrometers. Both quality issues appeared at site 5 and were probably caused by the moving of the instrumentation during lawn mowing.

The data from weather stations of MeteoSwiss and the airport are rather continuous and consistent.

The Appendix J presents the data availability of the devices during the campaign period.

**3  The database**

**3.1  Description of raw data**

The raw data files are stored in various ASCII formats in folders based on the device/weather station and are available at daily resolutions. The files are organised as depicted in Fig. 6. The filename format examples are presented for each folder.

### 3.2 Tool/HTML viewer

To facilitate the efficient exploration of the data plots, the html file makes it possible to readily plot pre-defined views of one selected day. The folders are related to the main data sources: the CML, the disdrometers, the rain gauges, the MeteoSwiss and airport weather stations. The intensity of red colour in the campaign calendar describes the daily cumulative rainfall depth, based on RG03, which enables the user to choose the most interesting days and explore them further. Once the day is selected, the plots in each folder are displayed.

There are eight pre-defined views in the drop-down menu in the viewer. The first view plots CML data accompanied by data from RG03 from site 2 which is located in the middle of the link path. Views two and three present rainfall intensities from the disdrometers, missing values and drop size distributions. The fourth view displays rain gauge data and its missing values. Views five and six display the data and missing values from the airport weather stations and the last two views concern the data and missing values from the MeteoSwiss weather stations. Note that November 2012 was extremely dry, therefore no rainfall was recorded.

Table 4 summarizes the daily plots of measured quantities. All plotted quantities are accompanied by a plot of missing values. Each column in those figures represents one hour of the day. The proportional amount of missing values in each hour is displayed in red.

## 4 Applications of the dataset and outlook

The following section contains a short review of important open issues related to rainfall estimation using microwave links for which the CoMMon field campaign provides relevant data and could give valuable input in future research.

### 4.1 Former applications

#### 4.1.1 Dry/wet classification and baseline estimation

Based on the dataset, Wang et al. (2012) developed a new algorithm for classifying dry/wet periods and estimating baseline attenuation using a Markov switching model which outperformed a previous method based on the standard deviation of link attenuation over a moving time window by Schleiss and Berne (2010). Using factor graph theory and robust local linear regression, Reller et al. (2011) and Schatzmann et al. (2012) used the CoMMon dataset to develop two alternative baseline models with similar offline/online performances.

#### 4.1.2 Wet antenna attenuation

Based on data collected during the CoMMon experiment, Schleiss et al. (2013) quantified the magnitude and dynamics of wet-antenna attenuation (WAA) affecting commercial microwave links at 38 GHz. They found WAA values in the order of 1-2 dB, with an upper bound at about 2.3 dB. Furthermore, WAA increased exponentially during the first 5-20 min of rain and decreased exponentially as soon as the rain stopped. Figure 7 presents an example of the attenuation pattern caused by a wet

**Table 4.** Summary of quantities and their units displayed as daily plots in the html viewer.

| | | |
|---|---|---|
| CML | Received power (x10) [dBm] | |
| | Transmitted power [dBm] | |
| Disdrometers | Rain rate [mm h$^{-1}$] | |
| | Cumulative rainfall [mm] | |
| | Drop diameter and fall velocity distribution | |
| | Temporal evolution of drop diameter | |
| Rain gauges | Rain rate [mm h$^{-1}$] | |
| | Cumulative rainfall [mm] | |
| MeteoSwiss weather stations | Sun radiation (10 min mean) [W m$^{-2}$] | |
| | Sunlight duration (10 min mean) [min] | |
| | Temperature (10 min) [°C] | |
| | Wind direction (10 min mean) [°] | |
| | Wind speed (10 min mean) [m s$^{-1}$] | |
| Airport weather stations | Dew point [°C] | |
| | Atmospheric pressure [hPa] | |
| | Relative humidity [%] | |
| | Rain intensity [mm h$^{-1}$] | |
| | Cumulative rainfall [mm] | |
| | Temperature [°C] | |
| | Wind direction (10 min mean) [°] | |
| | Wind speed (10 min mean) [m s$^{-1}$] | |

antenna. The rate at which WAA decreased after an event showed substantial variation, ranging from a few minutes to several

225 hours depending on temperature, wind and humidity. In a follow-up study, Fencl et al. (2014) assessed the effectiveness of direct antenna shielding for mitigating WAA compared with post-processing techniques. They found that antenna shielding helps to substantially reduce bias in rainfall estimates. However, shielding did not outperform model-based corrections as signals from shielded antennas still experienced attenuation, even when the face of the radome was completely dry. Whether this is caused by the attenuation of side-lobes or whether these are effects caused by changes in the scattering and reflection

behaviour of the surrounding buildings, such as nearby walls, roofs or impervious surfaces, during wet weather is currently unknown.

There are still several unresolved questions related to WAA, such as the effect of horizontal/vertical polarization on WAA magnitude or the quantification of WAA during fog and dew events (Fig. 8). For example, de Vos et al. (2019) showed that errors in CML quantitative precipitation estimates are the largest for observations during late night and early morning periods

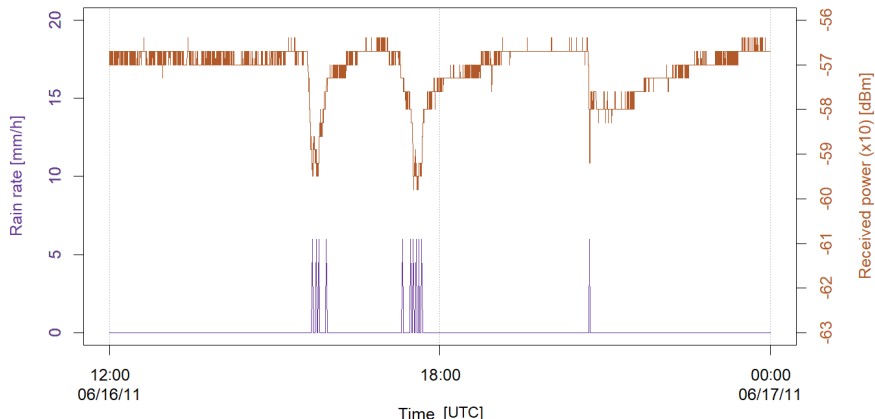

**Figure 7.** Wet antenna attenuation pattern. Rainfall causes signal resulting in residual wetness on the surface of the radomes. The return of baseline attenuation to previous dry-weather levels is attributed to the drying of the antenna radomes (Van Leth et al., 2018). The same behaviour was observed by new E-bands and reported recently in Fencl et al. (2020).

when dew is more likely to form on antennas. Van Leth et al. (2018) observed additional attenuation in the order of 3 dB during foggy weather conditions and whenever dew was present on the antennas. To be precise, Van Leth et al. (2018) explicitly mention that the theoretical upper limit for attenuation by fog droplets themselves is about 0.75 dB at 38 GHz for a 1 km path and a very high liquid water content of 0.8 g m$^{-3}$ . Therefore, no more than 1.5 dB km$^{-1}$ can be expected due to the fog itself and the rest is probably due WAA. So reasonable estimates of attenuation during heavy fog are in the order of 2-3 dB of WAA

+ 1-1.5 dB km$^{-1}$ for the fog itself. Heated antennas would be needed to prevent the formation of dew and/or air blowers would blow away the water during fog events. However, modelling these effects remains challenging, since the dew-related wetting depends on the weather conditions, as well as on the condition/presence of hydrophobic radome cover. The CoMMon dataset could help gain new insight into these issues, for example, by further investigating WAA due to dew formation on antenna radomes (Fig. 8).

### 4.1.3   DSD retrieval and DSD related errors

Attenuation data of CMLs operating at different frequencies or polarizations could be, in theory, used for estimating path-averaged raindrop size distributions (e.g., Rincon and Lang, 2002). Research on this issue is still ongoing. Recently, Song et al. (2019) used a simulation study to illustrate how to retrieve DSDs from dual-frequency dual-polarization links. Another study by van Leth et al. (2020) based on a similar approach showed that reasonable performance on selected events and idealized

conditions can be achieved. However, retrieved DSD parameters are not always reliable and large uncertainties remain due to quantization noise, baseline estimation and wet-antenna attenuation. These could possibly be reduced by using the data to update prior knowledge on empirical drop size distributions using Bayesian data analysis (Cao et al., 2010).

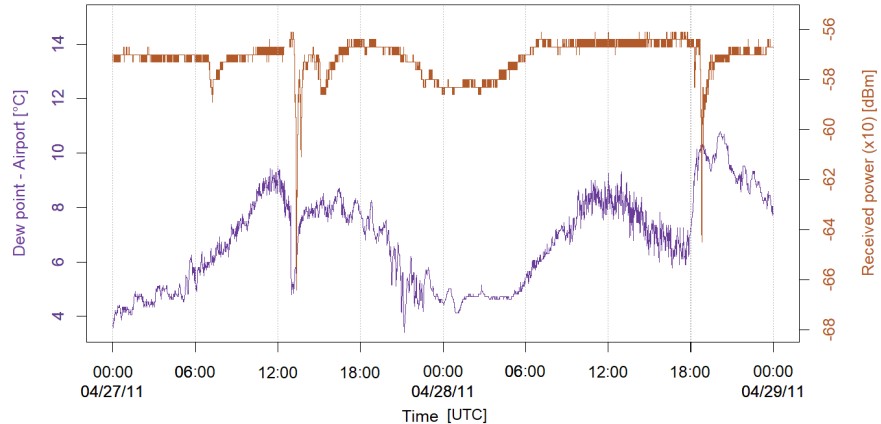

**Figure 8.** Wet antenna attenuation due to dew formation causes substantial attenuation from approx. 18:00 on 27 November 2011 to 7:00 on 28 November 2011.

Also, the high-quality CoMMon dataset could provide the evidence base to test different retrieval techniques and to assess their strengths and limitations.

The data will be useful to design outage-free radio communication systems better. Detailed data on rainfall microphysics and microwave attenuation from operational devices can be extremely contributive to radio engineers (Kvicera et al., 2009).

## 4.2   Outlook

There are still many unsolved issues regarding how to effectively retrieve precipitation from CML attenuation in praxis. Given the high temporal resolution, the CoMMon data might be most useful in improving our understanding of antenna wetting,
baseline dynamics and the impact of variable DSDs.

    First, wet antenna attenuation and dew formation on antennas are phenomena that need to be described more precisely to avoid the overestimation of rainfall. Several studies have suggested that correcting for wet antenna attenuation can significantly enhance results (Leijnse et al., 2008; Pastorek et al., submitted). Most probably, corrections cannot be based on frequency and signal dynamics only, since the atmospheric state around the CML varies. In other words, temperature, relative humidity,
radiation and wind probably have a more significant impact on the drying time of the antennas as well as the conditions prior to wetting. To what extent machine learning (Habi and Messer, 2018), trained on many CoMMon-type datasets, can provide an empirical solution remains to be seen.

    Second, the baseline of transmitted minus received power is often approximated as constant, even though substantial variations during dry weather have been reported (Wang et al., 2012) and there is little evidence that the baseline remains stationary
during wet periods. Different dry-wet weather classification approaches were presented in Schleiss and Berne (2010), Overeem et al. (2011) or Polz et al. (2020). A benchmarking activity with many datasets from different sites and climatic regions is still lacking.

Scattering theory suggests that a larger variation of drop size distribution challenges more precise retrieval for longer links (Leijnse et al., 2010). The disdrometer observations in the CoMMon dataset can also be used to build simulators making it possible to better understand the attenuation-rainfall relation and assess CML rainfall retrieval uncertainties related to variable DSDs (Berne and Uijlenhoet, 2007; Schleiss et al., 2012). For example the assumption that the path-integrated DSD can be represented by a simple unimodal distribution (e.g., a Gamma or exponential) might not be true for longer links, as rainfall rates and DSDs can exhibit significant heterogeneity along the path of the link. Similarly, for highly localized rain showers, only a small fraction of the link might actually experience rain and not all antennas might be equally wet/dry at a given time.

Variable DSDs also represent major uncertainties at CMLs with higher frequency bands. Although, to date, most CMLs use frequencies from 5 GHz to 40 GHz, the spectrum is currently further extended to 80 GHz. Recently, Fencl et al. (2020) used PARSIVEL observations from the CoMMon dataset to simulate rainfall retrieval from an E-band CML which demonstrated that these may be promising tools for sensing light rainfall which is challenging for lower frequencies due to the quantization of the attenuation data (Berne and Schleiss, 2009). In a similar fashion, CMLs are "blind" regarding extremely high intensities as attenuation due to such high intensities drops below the receiver threshold of the hardware and causes outages of the CML (cf. event on 5 August 2011). How this can be solved by "interpolating" missing observations based on signals from nearby sensors (Mital et al., 2020), remains to be seen.

Melting snow causes large attenuation of EM waves at frequencies commonly used by CMLs (ITU-R, 2015). Upton et al. (2007) identified periods with melting snow by analysing attenuation data of dual-frequency microwave links operating at 12.8 GHz/17.6 GHz and 10.5 GHz/17.5 GHz. Ostrometzky et al. (2015) suggested using CMLs operating at multiple frequencies to distinguish between periods with snow, melting snow and rainfall and thus improve estimation of total accumulated precipitation. Attenuation of EM waves by ice particles, however, remains challenging to simulate due to complex shapes of these hydrometeors. Moreover, ice particles containing liquid water interact with EM waves in substantially different manner (Eriksson, 2018).

# 5 Data accessibility

Data from CML, disdrometers and rain gauges, and nearby weather stations are available with data files stored in the Zenodo repository at https://doi.org/10.5281/zenodo.4923125 (Špačková et al., 2021). The citation should be used as follows:

- For the paper: Špačková, A., Bareš, V., Fencl, M., Schleiss, M., Jaffrain, J., Berne, A., and Rieckermann, J.: One year of attenuation data from a commercial dual-polarized duplex microwave link with concurrent disdrometer, rain gauge and weather observations, Earth Syst. Sci. Data, submitted.

- For the database: Špačková, A., Jaffrain, J., Wang, Z., Schleiss, M., Fencl, M., Bareš, V., Berne, A., and Rieckermann, J.: One year of attenuation data from a commercial dual-polarized duplex microwave link with concurrent disdrometer, rain gauge and weather observations, [Data set], Zenodo, https://doi.org/10.5281/zenodo.4923125, 2021.

The dataset is available for reuse under a CC BY 4.0 license. License terms apply.

## 6 Conclusions

The data from the CoMMon field campaign described in this paper is relevant for the remote sensing of rainfall, as well as for the design of outage-free terrestrial wireless communication systems. The unique dataset provides a comprehensive package of attenuation data from a 38 GHz dual-polarized CML with concurrent disdrometer and rain gauge measurements in (sub-)minute resolution. In addition, meteorological data from the weather stations of MeteoSwiss and Dübendorf airport were included. The main conclusions are:

- The remote sensing of precipitation and related atmospheric phenomena, such as dew, remains a relevant problem. Using signals from commercial telecommunication microwave links to learn about these phenomena seems promising because they cover sparsely or completely ungauged regions and can be queried remotely and fast. The open CoMMon dataset makes a unique contribution by providing dual-polarized transmitted and received power levels, as well as ground-level observations of precipitation microphysics and local weather. It fosters the interconnection of datasets which can be used to better understand scattering phenomena and benchmark retrieval methods.

- The dataset represents a duration of one year and contains data from i) a single 38-GHZ dual polarized CML with a length of 1.85 km; ii) collocated observations of five disdrometers; iii) three rain gauges; and iv) observations from five nearby weather stations. Specific highlights are, first, that the antenna radomes were protected by custom shielding for approximately half of the period of the campaign, thus making it possible to investigate the impact of antenna wetting which is still considered a major disturbance for rainfall retrieval. Second, the data are provided in sub-minute resolutions making it possible to investigate the detailed dynamics of the involved processes. Third, the dataset contains periods with rain but also periods during which ice hydrometeors including snow and melting snow occured (see Appendix I).

- Although the experimental campaign faced expected difficulties regarding sensor malfunctioning, data outages, etc., these episodes are well documented and, thus, do not compromise the usefulness of the dataset (Appendix J). The provided static HTML viewer also makes it easy to explore the data by pre-configured views of daily time series. For example, by focusing on days with intense or little precipitation, typical dynamics of the observed processes can be screened effortlessly.

- The dataset contains unique evidence regarding several processes such as the wetting and drying of antenna radomes and outdoor units or the impact of temperature and wind. We encourage several applications, from investigating baseline separation to wetting phenomena, such as dew, which had much slower dynamics in comparison to rain-induced attenuation, to the retrieval of drop-size distributions from the joint analysis of horizontal and vertical polarizations.

- In the future, the CoMMon dataset can be used to further investigate challenging issues in the remote sensing of rainfall, such as the classification of dry/wet periods, space-time variability of DSDs or even the analysis of fade margins for better radio network design.

# Appendix A: PARSIVEL drop diameter classes

**Table A1.** PARSIVEL drop diameter classes

| | Particle diameter classes | |
|:---:|:---:|:---:|
| **Class number** | **Class Average (mm)** | **Class Spread (mm)** |
| 1 | 0.062 | 0.125 |
| 2 | 0.187 | 0.125 |
| 3 | 0.312 | 0.125 |
| 4 | 0.437 | 0.125 |
| 5 | 0.562 | 0.125 |
| 6 | 0.687 | 0.125 |
| 7 | 0.812 | 0.125 |
| 8 | 0.937 | 0.125 |
| 9 | 1.062 | 0.125 |
| 10 | 1.187 | 0.125 |
| 11 | 1.375 | 0.250 |
| 12 | 1.625 | 0.250 |
| 13 | 1.875 | 0.250 |
| 14 | 2.125 | 0.250 |
| 15 | 2.375 | 0.250 |
| 16 | 2.750 | 0.500 |
| 17 | 3.250 | 0.500 |
| 18 | 3.750 | 0.500 |
| 19 | 4.250 | 0.500 |
| 20 | 4.750 | 0.500 |
| 21 | 5.500 | 1.000 |
| 22 | 6.500 | 1.000 |
| 23 | 7.500 | 1.000 |
| 24 | 8.500 | 1.000 |
| 25 | 9.500 | 1.000 |
| 26 | 11.000 | 2.000 |
| 27 | 13.000 | 2.000 |
| 28 | 15.000 | 2.000 |
| 29 | 17.000 | 2.000 |
| 30 | 19.000 | 2.000 |
| 31 | 21.500 | 3.000 |
| 32 | 24.500 | 3.000 |

## Appendix B: PARSIVEL drop velocity classes

**Table B1.** PARSIVEL drop velocity classes

| | Particle velocity classes | |
| :---: | :---: | :---: |
| **Class number** | **Class Average (m s$^{-1}$)** | **Class Spread (m s$^{-1}$)** |
| 1 | 0.050 | 0.100 |
| 2 | 0.150 | 0.100 |
| 3 | 0.250 | 0.100 |
| 4 | 0.350 | 0.100 |
| 5 | 0.450 | 0.100 |
| 6 | 0.550 | 0.100 |
| 7 | 0.650 | 0.100 |
| 8 | 0.750 | 0.100 |
| 9 | 0.850 | 0.100 |
| 10 | 0.950 | 0.100 |
| 11 | 1.100 | 0.200 |
| 12 | 1.300 | 0.200 |
| 13 | 1.500 | 0.200 |
| 14 | 1.700 | 0.200 |
| 15 | 1.900 | 0.200 |
| 16 | 2.200 | 0.400 |
| 17 | 2.600 | 0.400 |
| 18 | 3.000 | 0.400 |
| 19 | 3.400 | 0.400 |
| 20 | 3.800 | 0.400 |
| 21 | 4.400 | 0.800 |
| 22 | 5.200 | 0.800 |
| 23 | 6.000 | 0.800 |
| 24 | 6.800 | 0.800 |
| 25 | 7.600 | 0.800 |
| 26 | 8.800 | 1.600 |
| 27 | 10.400 | 1.600 |
| 28 | 12.000 | 1.600 |
| 29 | 13.600 | 1.600 |
| 30 | 15.200 | 1.600 |
| 31 | 17.600 | 3.200 |
| 32 | 20.800 | 3.200 |

# Appendix C: PARSIVEL raw data parameters

**Table C1.** PARSIVEL raw data parameters

| Position | Parameter | Format | Units |
|---|---|---|---|
| 1 | Date and Time | YYYY-MM-DD hh:mm:ss | UTC |
| 2 | Record number | - | - |
| 3 | Logger temperature | - | °C |
| 4 | Logger voltage | - | V |
| 5 | PARSIVEL rain rate | - | mm h$^{-1}$ |
| 6 | PARSIVEL rain amount | - | mm |
| 7 | Precipitation code 4680 | - | - |
| 8 | Precipitation code 4677 | - | - |
| 9 | PARSIVEL radar reflectivity | - | dBZ |
| 10 | Visibility in the precipitation | - | m |
| 11 | Laser amplitude | - | - |
| 12 | Number of particles detected | - | - |
| 13 | PARSIVEL temperature | - | °C |
| 14 | PARSIVEL heating current | - | A |
| 15 | PARSIVEL voltage | - | V |
| 16 | PARSIVEL status | - | - |
| 17 | Absolute amount | - | mm |
| 18 | Transmit time | - | - |
| 19 | Field N | Vector of 32 values | m$^{-3}$ mm$^{-1}$ |
| 20 | Field v | Vector of 32 values | m s$^{-1}$ |
| 21 | Raw data | Vector of 1024 values | - |
| 22 | Communication error | - | - |

Parameter 16 registers the status of the optical lenses (0 – everything OK, 1 – laser protective glass is dirty, but measurements are still possible, 2 – laser protective glass is dirty, partially covered, no further usable measurements are possible, 3 – laser damaged).

## Appendix D:  Precipitation code according to SYNOP

**Table D1.** Precipitation code according to SYNOP

| **No precipitation** | | Tab. 4680 | Tab. 4677 |
|---|---|---|---|
| | | 00 | 00 |

| **Drizzle** | | | |
|---|---|---|---|
| Intensity | Rain rate [mm/h] | Tab. 4680 | Tab. 4677 |
| light | < 0.1 | 51 | 51 |
| moderate | $\geq 0.1 \ldots < 0.5$ | 52 | 53 |
| heavy | $\geq 0.5$ | 53 | 55 |

| **Drizzle with rain** | | | |
|---|---|---|---|
| Intensity | Rain rate [mm/h] | Tab. 4680 | Tab. 4677 |
| light | < 2.5 | 57 | 58 |
| moderate | $\geq 2.5 \ldots < 10.0$ | 58 | 59 |
| heavy | $\geq 10.0$ | 58 | 59 |

| **Rain** | | | |
|---|---|---|---|
| Intensity | Rain rate [mm/h] | Tab. 4680 | Tab. 4677 |
| light | < 2.5 | 61 | 61 |
| moderate | $\geq 2.5 \ldots < 10.0$ | 62 | 63 |
| heavy | $\geq 10.0$ | 63 | 65 |

| **Rain, drizzle with snow** | | | |
|---|---|---|---|
| Intensity | Rain rate [mm/h] | Tab. 4680 | Tab. 4677 |
| light | < 2.5 | 67 | 68 |
| moderate | $\geq 2.5 \ldots < 10.0$ | 68 | 69 |
| heavy | $\geq 10.0$ | 68 | 69 |

| **Snow** | | | |
|---|---|---|---|
| Intensity | Rain rate [mm/h] | Tab. 4680 | Tab. 4677 |
| light | < 1.0 | 71 | 71 |
| moderate | $\geq 1.0 \ldots < 4.0$ | 72 | 73 |
| heavy | $\geq 4.0$ | 73 | 75 |

| **Snow grains** | | | |
|---|---|---|---|
| Intensity | Rain rate [mm/h] | Tab. 4680 | Tab. 4677 |
| —[1] | > 0 | 77 | 77 |

| **Soft hail** | | | |
|---|---|---|---|
| Intensity | Rain rate [mm/h] | Tab. 4680 | Tab. 4677 |
| light | < 1.0 | 87 | 87 |
| mod./heavy | $\geq 1.0$ | 88 | 88 |

| **Hail** | | | |
|---|---|---|---|
| Intensity | Rain rate [mm/h] | Tab. 4680 | Tab. 4677 |
| light | < 2.5 | 89 | 89 |
| mod./heavy | $\geq 2.5$ | 89 | 90 |

[1] no classification made

# 340    Appendix E:  CML: measured parameters

**Table E1.** CML: measured parameters

| Column no. | Parameter | Units |
|:---:|:---:|:---:|
| 1 | Date and time, format YYYY-MM-DD hh:mm:ss | UTC |
| 2 | Tx W $\longrightarrow$ D, H | dBm |
| 3 | Rx x10 W $\longrightarrow$ D, H | dBm |
| 4 | Tx W $\longrightarrow$ D, V | dBm |
| 5 | Rx x10 W $\longrightarrow$ D, V | dBm |
| 6 | Tx D $\longrightarrow$ W, H | dBm |
| 7 | Rx x10 D $\longrightarrow$ W, H | dBm |
| 8 | Tx D $\longrightarrow$ W, V | dBm |
| 9 | Rx x10 D $\longrightarrow$ W, V | dBm |

Transmitted power (Tx); Received power (Rx); Horizontal polarization (H); Vertical polarization
(V); site 5 - Wangen (W); site 1 - Dübendorf (D); $\longrightarrow$ direction of the signal

# Appendix F:  Rain gauges: measured parameters

**Table F1.** Rain gauges: measured parameters

| Column no. | Parameter | Units |
|:---:|:---:|:---:|
| 1 | Date and time, format YYYY-MM-DD hh:mm:ss | UTC |
| 2 | Number of tips per time step | - |
| 3 | Rain rate | mm h$^{-1}$ |
| 4 | Rain amount per time step | mm |
| 5 | Cumulative rain amount | mm |

## Appendix G:  MeteoSwiss weather station parameters

**Table G1.** MeteoSwiss weather station parameters

| Column no. | MeteoSwiss name | Description | Measurement sampling |
|:---:|:---:|:---:|:---:|
| 1 | STN | MeteoSwiss station number | |
| 2 | time | Measurement time, format YYYY-MM-DD hh:mm:ss | |
| 3 | tre200s0 | Air temperature 2 m above ground | Instantaneous (10 min resolution) |
| 4 | tko200ax | Air temperature 2 m above ground | Half-day max |
| 5 | tko200an | Air temperature 2 m above ground | Half-day min |
| 6 | prestas0 | Air pressure at the height of the station | Instantaneous (10 min resolution) |
| 7 | fkl010z1 | Gust wind speed | Maximum |
| 8 | rre150z0 | Precipitation | 10-min sum |
| 9 | rco150z0 | Precipitation duration | 10-min sum |
| 10 | sre000z0 | Sunlight duration | 10-min sum |
| 11 | dkl010z0 | Wind direction | 10-min mean |
| 12 | fkl010z0 | Wind speed | 10-min mean |
| 13 | pp0qs0 | Air pressure at sea level | Instantaneous (10 min resolution) |
| 14 | pp0qnhs0 | Air pressure at sea level in standard atmosphere | Instantaneous (10 min resolution) |
| 15 | gre000z0 | Sun radiation | 10-min mean |
| 16 | tre005s0 | Air temperature at 5 cm above grass | Instantaneous (10 min resolution) |
| 17 | tde200s0 | Dew point at 2 m above ground | Instantaneous (10 min resolution) |

## Appendix H:  Airport weather station parameters

**Table H1.** Airport weather station parameters

| Column no. | Parameter | Units |
|:---:|:---:|:---:|
| 1 | Date and time, format YYYY-MM-DD hh:mm:ss | UTC |
| 2 | Temperature | °C |
| 3 | Dew point | °C |
| 4 | Relative humidity | % |
| 5 | Pressure | hPa |
| 6 | Rain intensity | mm h$^{-1}$ |
| 7 | Wind speed | m s$^{-1}$ |
| 8 | Wind direction | ° |

# Appendix I: Table of precipitation events

**Table I1.** Example of precipitation event table. Complete table can be found in the Zenodo repository.

| | Starting time [GMT] | Ending time [GMT] | Parsivels 20 | 21 | 22 | 40 | 41 | Peak mean R [mm h⁻¹] | Mean amount [mm] | Max R [mm h⁻¹] | Sta max R | Min Ra [mm] | Sta min Ra | Max Ra [mm] | Sta max Ra | Predominant Prec. type (%) |
|---|---|---|---|---|---|---|---|---|---|---|---|---|---|---|---|---|
| 1 | 17.03.2011 14:06:00 | 17.03.2011 18:46:30 | x | x | x | - | - | 4,11 | 3,72 | 5,22 | 21 | 3,25 | 20 | 4,38 | 21 | Rain (100%) |
| 2 | 19.03.2011 00:03:30 | 19.03.2011 16:20:00 | x | x | x | - | - | 5,29 | 12,17 | 6,50 | 20 | 6,38 | 40 | 14,43 | 21 | Rain (99.8%) |
| 3 | 28.03.2011 05:14:30 | 28.03.2011 10:54:30 | x | - | x | x | - | 3,95 | 3,75 | 7,95 | 21 | 3,18 | 40 | 4,46 | 21 | Rain (99.3%) |
| 4 | 31.03.2011 11:42:00 | 31.03.2011 17:07:30 | - | - | - | x | - | 6,39 | 4,43 | 6,39 | 40 | 4,39 | 40 | 4,39 | 40 | Rain (100%) |
| 5 | 04.04.2011 02:44:00 | 04.04.2011 09:51:00 | x | x | x | x | - | 9,42 | 16,87 | 15,62 | 40 | 14,97 | 20 | 19,14 | 21 | Rain (99.8%) |
| 6 | 04.04.2011 13:31:30 | 04.04.2011 15:30:00 | x | x | x | x | x | 3,62 | 2,00 | 10,96 | 22 | 1,32 | 20 | 2,82 | 21 | Rain (100%) |
| 7 | 27.04.2011 12:57:30 | 27.04.2011 15:42:30 | x | x | x | x | x | 31,50 | 3,70 | 72,52 | 21 | 2,74 | 20 | 5,05 | 20 | Rain (97.9%) |
| 8 | 28.04.2011 18:00:00 | 28.04.2011 19:55:00 | x | x | x | x | x | 15,34 | 2,06 | 35,32 | 40 | 0,70 | 40 | 3,21 | 40 | Rain (99.4%) |
| 9 | 03.05.2011 03:42:00 | 03.05.2011 07:02:30 | x | x | x | x | x | 21,62 | 7,43 | 39,87 | 21 | 6,80 | 40 | 8,90 | 21 | Rain (98.7%) |
| 10 | 11.05.2011 17:06:30 | 11.05.2011 18:16:00 | x | x | x | x | x | 19,73 | 2,83 | 43,20 | 21 | 1,30 | 20 | 3,77 | 41 | Rain (100%) |
| 11 | 12.05.2011 07:41:00 | 12.05.2011 10:27:30 | - | x | - | x | x | 19,45 | 5,19 | 53,20 | 21 | 0,08 | 22 | 7,13 | 21 | Rain (99.6%) |
| 12 | 12.05.2011 14:24:00 | 12.05.2011 18:14:30 | x | x | x | x | x | 57,53 | 13,82 | 80,91 | 20 | 11,76 | 22 | 16,73 | 21 | Rain (96.7%) |
| 13 | 14.05.2011 12:50:30 | 14.05.2011 14:47:00 | x | x | x | x | x | 3,63 | 1,03 | 4,82 | 22 | 0,87 | 20 | 1,17 | 21 | Rain (100%) |
| 14 | 14.05.2011 14:49:00 | 15.05.2011 02:02:30 | x | x | x | x | x | 5,48 | 15,53 | 10,56 | 40 | 13,53 | 20 | 16,70 | 21 | Rain (99.9%) |
| 15 | 15.05.2011 12:01:30 | 15.05.2011 13:20:30 | x | x | x | x | x | 10,83 | 3,33 | 35,22 | 20 | 2,30 | 40 | 4,45 | 20 | Rain (91.2%) |
| 16 | 15.05.2011 15:10:30 | 15.05.2011 16:17:30 | x | x | x | x | x | 10,16 | 3,30 | 14,78 | 21 | 2,74 | 40 | 3,79 | 21 | Rain (98.9%) |
| ⋮ | ⋮ | ⋮ | ⋮ | ⋮ | ⋮ | ⋮ | ⋮ | ⋮ | ⋮ | ⋮ | ⋮ | ⋮ | ⋮ | ⋮ | ⋮ | ⋮ |

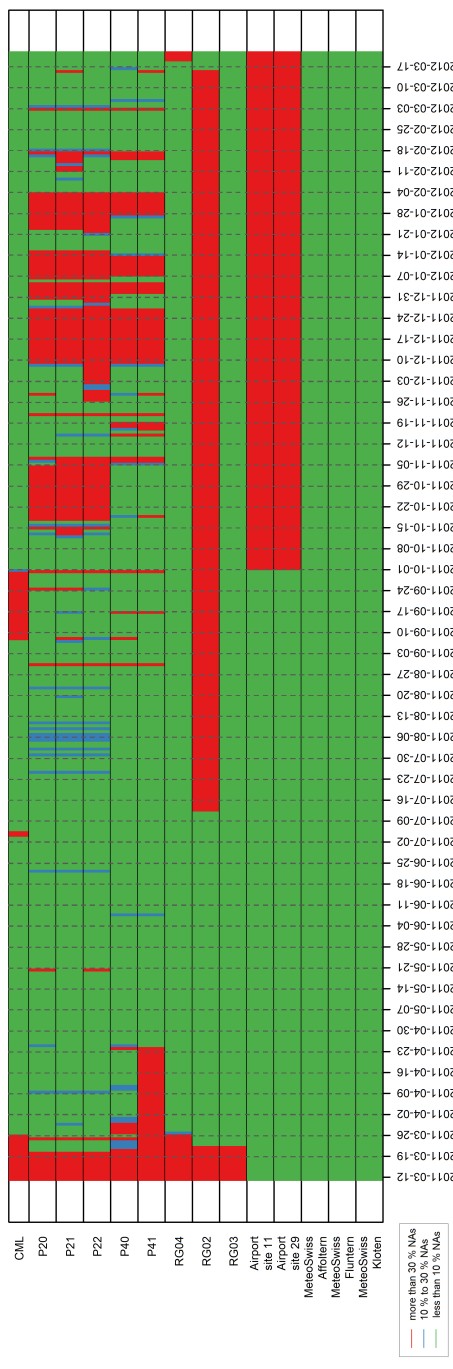

**Figure J1.** Data availability scheme. Red: more than 30 % NAs. Blue: 10 % to 30 % NAs. Green: less than 10 % NAs.

*Author contributions.* AŠ processed the data, with support from JJ, plotted the results and wrote the manuscript with support from MS, JR, MF and VB. The section regarding the applications of the dataset and outlook was written by MS. JJ and AB provided valuable feedback on the manuscript. JJ, MS, ZW, JR and AB designed the experimental layout and installed and operated the CML, rain gauges and disdrometers. AB and JR conceived the study and were in charge of overall direction and planning.

*Competing interests.* The authors declare that they have no conflict of interest.

*Acknowledgements.* The authors greatly acknowledge financial support from the Eawag and the Czech Science Foundation (GACR) project SpraiLINK 20-14151J. We thank all the persons who helped to perform the campaign. We want to mention Zhe Wang, Tobias Doppler, Richard Fankhauser, Vahab Rostampour (Eawag), Paul Stump, Angelo di Boni (RUAG), Christoph Wirz (Dübendorf airport) and Andre Studzinski, (EPFL). We thank Martin Kryl for support with the implementation of the html viewer.

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
