# Peer review of "One year of attenuation data from a commercial dual-polarized duplex microwave link with concurrent disdrometer, rain gauge and weather observations"

_Earth System Science Data, 2021_

## Author Comment (AC1)

**Responses to comments posted by Referee #1**

We thank the first referee for reviewing our article and providing beneficial feedback. Several unclear issues were clarified and significantly helped to improve the presentation of the campaign. We answer all comments in the following text. Our answers are in **bold**.

General comments

- Chapter four is not consistent in its content: While 4.1 does only give a summary of past activities on dry/wet classification with the presented data, 4.2 and 4.3 also include open issues with regard to wet antenna attenuation and DSD estimation. In 4.4 the issues from the preceding sub-sections are (partly) repeated with less structure and additional issues are presented. The whole section 4 should be improved by restructuring its content - for example in sub-sections covering the already conducted applications of the data set and the unsolved issues which could be tackled in the future - in a consistently manner.

**Chapter 4 has been restructured into two main subsections. First subsection (4.1 Former application) deals with the past applications of the data set and contains three separate parts (4.1.1 Dry/wet classification and baseline estimation, 4.1.2 Wet antenna attenuation and 4.1.3 DSD retrieval and DSD related errors). The second subsection (4.2 Outlook) is devoted to the outlook.**

- For quick exploration and convenience of the data user a global overview of the data availability and data issues (flags) describe in section 2.5 should be added both to the manuscript and the event calendar view in the html viewer. This would be most important for the CML data and could be done e.g. with an additional figure in the manuscript showing time vs data availability/flags and likewise or with and an additional calendar view in the html viewer. The daily data availability charts are fine, but it takes rather long time to select e.g. the "best" summer month for a certain application. (see Fig. 2 in van Leth et al., 2018)

**The figure was added to the appendix of the manuscript and in the html viewer.**

[Figure]

- An additional kml file of the observation locations in the data publication would add further convenience for the user of the data.

**The kml file was added to the Zenodo repository.**

Detailed comments:

p.1 l.2f.: Here you show CML use cases which are not further described in the text. I suggest to either give a short overview of the retrieval of such variables and the potential of the presented data set or to concentrate on rainfall as target variable as you did consistently in the manuscript

**The second sentence was changed. Now we do not address the issue of all presented cases but focus mainly on the rainfall.**

**The changes in the original text are in red:**

**Commercial microwave links (CML) in telecommunication networks can provide relevant information for remote sensing of precipitation and other environmental variables, such as path-averaged drop size distribution, evaporation, or humidity. The CoMMon field experiment (COmmercial Microwave links for urban rainfall MONitoring) mainly focused on the rainfall observations by monitor ing a 38-GHz dual-polarized CML of 1.85 km path length at a high temporal resolution (4 s), as well as a collocated array of five 5 disdrometers and three rain gauges over one year.**

p1. l.15: delete "also"

**Deleted.**

p2. l.20: Use CML attenuation rather than microwave attenuation to be more consistent (e.g. with p.2 l.25).

**"microwave attenuation" was replaced by "CML attenuation" in p2. l.20**

**"Attenuation data of microwave links" was replaced by "Attenuation data of CMLs" in p12. l.204**

**"microwave attenuation" was replaced by "CML attenuation" in p14. l.216**

p.2 l.25ff.: Refer to the availability of open source packages which for CML processing accompanied by small test data sets (e.g. https://github.com/overeem11/RAINLINK and https://github.com/pycomlink/pycomlink).

Sentence "Nevertheless, there are openly available tools for CML rainfall information processing and mapping (e.g., Overeem et al., 2016; Chwala et al., 2020)." was added to p.2 l.27.

**Corresponding citations were added to the references.**

**Overeem, A., Leijnse, H., and Uijlenhoet, R.: Retrieval algorithm for rainfall mapping from microwave links in a cellular communication network, Atmos. Meas. Tech., 9, 2425–2444, https://doi.org/10.5194/amt-9-2425-2016, 2016.**

**Chwala, C., Keis, F., Graf, M., Sereb, D., and Boose, Y.: pycomlink software package, v0.2.5, available at: https://github.com/pycomlink/pycomlink, last access 6 March 2021, 2020.**

p.3 l.57f.: Add citation stating that parameter b is close to 1 for a certain frequency range (e.g from ITU).

**The following citation was added in the text and the references.**

**ITU-R: ITU-R P.838-3, available at: http://www.itu.int/dms_pubrec/itu-r/rec/p/R-REC-P.838-3-200503-I!!PDF-E.pdf, last access: 6 March 2021, 2005.**

p.4 Figure 1: Use different colors for the zooming window and the CML. You also could plot the CML in the un-zoomed map without the disdrometers etc. for a better overview.

**The colour of the CML was changed to blue and the un-zoomed link path was added.**

[Figure]

p.4 Caption Figure 1: The Figure caption states "the direction of the link" please change to "the path of the CML" and "are less than 6 to 10 km" to "are within 6 to 10 km".

**Both phrases were changed. The changes in the original caption text are in red:**

**… The  path of the CML is in  blue ...**

**… The MeteoSwiss weather stations (yellow pins) are  within 6 to 10 km ...**

p.4 l.90: Be more precise about the quantization of the received signal

**The changes in the original text are in red:**

**The resolution of the transmitted power (Tx) was 1 dBm and for received power (Rx) it was  1/3 dBm expressed to one decimal place.**

p.5 l.95: Make clear that even with a shield wet antenna attenuation can potentially be a problem due to dew

**The changes in the original text are in red:**

**During the measurement campaign, plastic shields were installed on the antennas on 6 October 2011 to avoid water films on the antenna radomes and to eliminate wet antenna attenuation due to rainfall. However, the wet antenna attenuation due to dew cannot be prevented.**

p.5 l.108ff: Use plural or singular for rain gauge(s) consistently.

**The changes in the original text are in red:**

**Deployed 50 cm from the ground,  they had a sampling area of 400 cm$^2$.  Their bucket content of 4 g corresponded to the resolution of 0.1 mm of rain. The logger s had a time resolution 0.1 s and  their time drift was less than 2 min per month.**

p.6 l.117: Rephrase to "CML attenuation data is available for the period between …".

**Rephrased to: "CML attenuation data is available for the period between..."**

p.7 l.119: State again that the reading is instantaneous.

**The changes in the original text are in red:**

**The sampling interval of the instantaneous readings was 4 s.**

p.9 Figure 5: You could identify the individual rain gauges and disdrometers with small labels. For the caption you could move the data availability etc. to the running text.

**The labels of individual rain gauges and disdrometers were added. The caption regarding the data availability is already covered also in the running text. The description of the device outages is a substantial part for understanding the graph, therefore a short link to that description in the running text was written**

**The changes in the original text are in** red**:**

**Figure 5. Comparison of cumulative rain of the three rain gauges and five disdrometers. There were recorded several outages, suspicious measurements and technical issues, such periods are further described in the Data quality and reliability chapter.** ~~The operation of rain gauge was interrupted in July 2011. Another rain gauge was partially blocked from the middle of June 2011 until the 6 July 2011. The same rain gauge recorded unrealistic rainfall in August 2011. The outages of disdrometers between December and February caused an underestimation of rain amounts in this period. Disdrometer P22 recorded unrealistic rain amounts on 7 and 8 February 2012.~~ **Despite this, both types of devices are, in general, in good agreement.**

[Figure]

p.10 l. 164f: Here as well as in the html viewer it is unclear from which observation the daily cumulative rainfall depth is used. Please provide this information in the manuscript as well as in the html viewer.

**The changes in the original text are in** red**:**

**The intensity of red colour in the campaign calendar describes the daily cumulative rainfall depth based on RG03, which enables the user to choose the most interesting days and explore them further.**

**This information was added also to the html viewer.**

p 14. l.213: Delete "last, but not least".

**Deleted.**

---

## Author Comment (AC2)

**Responses to comments posted by Referee #2**

We thank the second referee for reviewing our article and providing beneficial feedback with many constructive thoughts, they significantly helped to improve the presentation of the measurement campaign. We answer all comments in the following text. Our answers are in **bold**.

In this paper, the author presents a dataset of microwave signal attenuation of a 38-GHz dual-polarized commercial microwave link (CML). The dataset was collected in the context of a one-year-long field campaign and is supplemented with the data of five laser optical disdrometers and three standard tipping bucket rain gauges under the path of the microwave link. In addition, further meteorological data from five synoptic weather stations is given.

General comments

As a whole would rate the datapaper as good and support to publish it with some revisions.

The dataset by itself is definitely unique and can serve as a basis for further studies on different issues dealing with quantitative precipitation estimation with the use of CML. On the one hand it is the availability at all, since CML data is often not openly a not available. On the other hand, it is the extend of the data. This holds true for the long-time span of the dataset (one year) of the CML data and also for the collocated data. Next to standard rain gauge and further (synoptic) meteorological data (temperature, humidity, wind and pressure), information of the real drop size distributions is given via the output of five laser optical disdrometers. To sum it up: with this dataset further detailed studies in the evolving field of QPE using CML are made possible.

The presentation of the dataset is generally good. The structure is clear and the data is easy to access. It is possible to find the data one is looking for, to download the data and to work with the data. The data format and the internal structure within the specific datasets is straight. Further on an overview is given by a HTML viewer, which is also easy to access and leads – via the color coding of the different dates based on the precipitation amount - to the – for the majority of the users - more interesting days with higher precipitation amounts.

Nevertheless, there are some – in my opinion – important points, which need a deeper revision.

The two most important point deals with (a) the metadata of the different sensors and stations and (b) a deeper discussion of the data quality. The metadata being one of the most important parts of a dataset, since they enables the future user to get a bit more insights to the situation the data was gathered or – to put it in other words – to comprehend the whole story a bit more. "*Metadata are especially important for elements which are particularly sensitive to exposure, such as precipitation, wind and temperature*" (Burt, 2012). In this paper and dataset, the metadata, as well as the discussion of the different data quality issues come – in my opinion – a bit very short. I couldn't find detailed information about the different measurement sites. Generally: For every site an own somehow detailed metadata-sheet (as recommended by the WMO and is the best practice in meteorological measurements) is needed, which should – next to a description of the site and the exposure, the specific instruments in use- also include

a couple of photos and a more detailed map. I also would recommend a maintenance plan with information what was done during maintenance.

**Metadata sheets were created and uploaded to the Zenodo repository. Information from maintenance reports was added to the running text (in section Data quality and reliability) and is also in the metadata sheets of each device. Information about the metadata sheets was also added to the running text (in section Instrumentation). Unfortunately, we were not able to get more detailed information about two airport weather stations, which provide additional meteorological data for the first seven months of the campaign, therefore no metadata sheets were created for them.**

**The changes in the original text in section Data quality and reliability are in red:**

**Major field visits were conducted on 15 April 2011, 6 July 2011, 19 October 2011, 1 November 2011 and 16<s>5</s> March 2012 <s>and 21 March 2012</s> to maintain the instruments. On 15 April 2011, the minor levelling corrections and data collections for rain gauges were done. On 6 July 2011 were batteries replaced and data collected. On 19 October 2011 was visually inspected RG04 with no recorded problems. On 1 November 2011 was RG02 and RG04 visually inspected with no reported issues. On 16 of March 2012, the RG02 was found blocked by a leaf and with empty batteries. The RG02 lagged laptop time by approximately 7 minutes. The difference between RG03 time and UTC was -00:05:31. The difference for RG04 was -00:02:28. The rain gauges were dynamically calibrated...**

In addition - regarding the different sensors / instruments:

For the disdrometers it would be important to know to which direction each was aligned to, since – because of the construction of the Parsivel laser optical disdrometers precipitation coming from the side is a bit more undisturbed. A short discussion of each disdrometer site with respect to possible wind influences would come handy as well as information of a change of the surrounding (especially trees and bushes) over the course of the year.

**The disdrometer positions and their surrounding information were added to the running text (in section Instrumentation) and metadata sheets.**

**The changes in the original text are in red:**

**...The horizontal laser beam had an area of 54 cm$^2$. The beam was oriented perpendicularly to the dominant wind direction in order to limit undersampling and splashing. The devices were deployed as far as possible from possible sources of turbulence and wakes (roof edges and other obstacles). The cob webs were not specifically prevented and during the field visits were not observed. The measurement principle...**

For the rain gauges a short discussion on possible wind influences and possible change of the surroundings is also recommended warmly.

**The rain gauge surrounding information and possible wind influences were added to the running text (in section Instrumentation) and metadata sheets.**

**The changes in the original text are in red:**

**...dynamically calibrated (Humphrey et al., 1997).** **The devices were deployed as far as possible from possible sources of turbulence and wake (roof edges and other structures).** **Deployed 50 cm from the ground** **(35 cm of the device itself and 15 cm of piece of wood and concrete)****,**  **they had a sampling area …**

For the CML the information about the height above the ground is missing and of high importance for the use of the data for further studies.

**The antennas' height above ground was added to the running text (in section Instrumentation as recommended in detailed comment p.4 l. 87 ff below). Moreover, the longitudinal terrain profile of the link path was added to Figure 1 with information about the antennas' height above ground.**

For the airport weather stations and meteo swiss weather stations a short description (list) of the different devices (sensor type, manufacturer, model) is needed in order to generally reproduce the measurements and gain information about the measuring principle. For the airport weather stations it is also important to know whether the international measuring standards are met.

**Matadata sheets of MeteoSwiss weather stations were created. Airport weather stations are maintained by professionals and the international measuring standards are met. Unfortunately, we were not able to get more detailed information about the two airport weather stations, which provide additional meteorological data for the first seven months of the campaign, therefore no metadata sheets were created for them.**

The second point which needs a more detailed description, and is so far a bit short.

This point goes a little bit hand in hand with the more detailed metadata and is partly accounted for, when one puts down more detailed metadata (for example a short discussion of the exposure of each site). But one further point is the need to clearly explain how often the instruments were maintained and what was done during maintenance. For the disdrometers it is for example interesting to know how possible cob webs were dealt with. For the rain gauge it is important to know, whether wind protection shields were used, or what was done during snow.

**The maintenance history of the devices was described in metadata sheets and specific issues regarding the cob webs and winter measurements were clarified.**

**The changes in the original text (section Instrumentation) are in red:**

**...The horizontal laser beam had an area of 54 cm². The beam was oriented perpendicularly to the dominant wind direction in order to limit undersampling and splashing. The devices were deployed as far as possible from possible sources of turbulence and wake (roof edges and other obstacles). The cob webs were not specifically prevented and during the field visits were not observed. The measurement principle …**

**...corresponded to the resolution of 0.1 mm of rain. The rain gauges were equipped neither with wind protection shields nor with heating, but only 6 snow and 4 mixed precipitation events were observed during the campaign. The loggers had a time resolution 0.1 s...**

Burt, S., 2012: *The weather observer's handbook,* Cambridge University Press, Cambridge, 444 p.

Detailed comments:

p.1 l.4: … CML of 1.85 km at a high temporal resolution  --- CML of 1.85 km path length at a high …

**Changed.**

p.1 l.6:   delete 'effortlessly'

**Deleted.**

p.1. l. 7: satisfactory to what? Be a bit more precise. Especially state, whether it is reliable.

**The changes in the original text are in red:**

**The data quality is generally satisfactory to the further analysis and potentially problematic measurements are flagged to help the analyst identify relevant periods for specific study purposes. The analyses conducted so far suggest that the data are reliable and of satisfactory quality.**

p.3. l.66: Formula (4): Give a reference, where the formula comes from. There are some little bit different moments of the DSD for the rain rate circulating in the literature.

**The changes in the original text are in red:**

**...The accuracy of the power-law approximation (Eq. 2) can be assessed by comparing Eq. 3 to the rain rate R (Eq. 4) of the observed drop spectrum (Atlas and Ulbrich, 1977)...**

1. 3 l. 73: Five laser optical disdrometers.

**Changed.**

p.3 l. 76. 'integrated', this verb seems in this context a bit not the best expression. Just write that three tipping bucket rain gauges were used.

**The changes in the original text are in red:**

**...For the campaign were also  used three tipping bucket rain gauges...**

p.4. Figure 1: Give an explanation to the abbreviations: P and RG.

**The changes in the original text are in red:**

**...Disdrometer (P - Parsivel) and rain gauge (RG - rain gauge) positions are indicated by white pins. …**

p.4 l. 86 ff: Section 2.3. use a subsection for each instrument and – as stated – give a link to the specific metadata sheets

**Subsections for each instrument were used. Information about the metadata sheets was added to the running text in each subsection.**

**New subsections are following:**

**2.3.1 Commercial microwave link**

**2.3.2 Laser optical disdrometers**

**2.3.3 Tipping bucket rain gauges**

**2.3.4 Weather stations**

p.4 l. 87 ff: CML- as already written – the height above ground is a needed information.

**The changes in the original text are in red:**

**...The length of the link was 1850 m. The antenna at site 1 was deployed 6 m above ground and the antenna at site 5 was deployed 5 m above ground. Measurement intervals…**

**Figure 1 was extended, subfigure 1b provides a longitudinal terrain profile corresponding to the path of the CML.**

**The changes in the original title are in red:**

**Figure 1. a) Layout of the CoMMon field campaign. The radio units are deployed at the end sites (site 1 and site 5). The  path of the CML is in  blue. Disdrometer (P - Parsivel) and rain gauge (RG - rain gauge) positions are indicated by white pins. Site 2 contained two collocated disdrometers for data quality control. Except for site 3, all rain gauges have a collocated disdrometer. The MeteoSwiss weather stations (yellow pins) are  within 6 to 10 km . Two additional weather stations (green) are located at the Dübendorf airport (site 11 and site 29). b) Longitudinal profile of terrain corresponding to the CML path.**

[Figure]

[Figure]

5. 5. l. 96. – laser optical disdrometer

**Changed.**

p.6. l. 1. Table 1: I think it is just the location of the sites, which is stated, not the characteristics

**The changes in the original title are in red:**

**Table 1.  Locations of the sites and deployed devices.**

6. 6. l.111 ff 'located 6 to 10 km from the experimental site' Be more precise, give the exact distances and directions.

**The changes in the original text are in red:**

**An additional three weather stations operated by MeteoSwiss provided comprehensive weather data at  stations located  8.9 km and azimuth 285° (Affoltern), 5.9 km and azimuth 238° (Fluntern), 10.9 km and azimuth 318° (Kloten) from the experimental site 3 (Fig. 1a).**

7. 6. l. 116 ff think this is a good place to state that the timestamp is UTC (not just in the Appendix).

**The following sentence was added to the end of the subsection 2.4 Measured variables:**

**"All measured variables are recorded in UTC time format."**

8. 7. l. 121 ff: in the Parsivel dataset there are 20 variables. Some (esp. "Precipitation code 4680", "Precipitation code 4677") need more explanation and in the "read me file" no units are given. I think this should be changed.

**… The eight types of precipitation (parameter 07 and 08 correspond to precipitation codes according to SYNOP) were classified based on a velocity-diameter relationship (Löffler-Mang and Joss, 2000). The code corresponds to the specific precipitation type and its intensity (OTT, 2006) …**

**Following reference was added:**

**OTT, Operating instructions: Present Weather Sensor Parsivel, 70.200.005.B.E 08-1008, 2006.**

**Units for disdrometer variables were added to read me file.**

p.8. Figure 3. X axis label / time stamp label should be time of the day and there should be a tie stamp (UTC) – This concerns also the other figures and especially the figures accessed via the html viewer.

**The UTC was added to the time axis and the time labels were enhanced with the date information.**

**The changes in the description in the html viewer are in red:**

**HMTL viewer serves for quick preview of available data from the experiment in the form of daily time series. It includes key observations collected during the experimental period including the information about data availaility. The presented plots are in UTC time format. The calendar provides access to daily data. …**

8.  8. l. 135 ff: 'all major weather characteristics' is absolutely not the right term. Better use for example standard meteorological parameters.

**The changes in the original text are in red:**

**The airport weather station data, collected for the time period from 1 March 2011 to 27 September 2011, contains  monitored standard meteorological parameters such as temperature, dew point, relative humidity, pressure, rain intensity, wind speed and direction with columns organised as described in Appendix G.**

9.  8. l. 137: As stated in the general comment section: The discussion of the data quality is partly a bit too common. What about the standard issues of rain gauge and disdrometer measurements regarding the wind influence and dirt ( and cob webs) and snow.

**Similarly as in section Instrumentation, the information about possible sources of negative influences was added.**

**The changes in the original text are in red:**

**… The filter presented in Jaffrain and Berne (2011) can be used to remove dubious measurements. Disdrometer locations and orientations were chosen to minimize wind disturbances, staying as far away from sharp corners and tall objects as possible. Cob webs were not observed during the field visits.**

**The rain gauges were deployed in adequate distance from trees and bushes to minimize the influence of wind whirling and clogging by leaves. However, One of the tipping bucket rain gauges faced technical issues that constrained the data collection. The greatest data availability gap, due to low batteries, happened at site 4 (RG02) between 6 July 2011 and 16 March 2012. ...**

p.9. l.142 ff. There is somehow a duplication of the description, please check this. I was a bit confused.

**The changes in the original text are in red:**

**All disdrometers partially malfunctioned between 7 December 2011 and 14 January 2012; and between 26 January 2012 and 3 February 2012 because of power shortage.**

11. 11. Table 4. Be a bit more precise in the label of the table.

**The changes in the original title are in red:**

**Table 4. Summary of  quantities and their units displayed as daily plots in the html viewer.**

12. 12. l. 178: has helped develop or could provide – a little bit confusing sentence.

**The changes in the original text are in red:**

**The following section contains a short review of important open issues related to rainfall estimation using microwave links for which the CoMMon field campaign  provides relevant data and could  give valuable input in future research.**

p.12. l.179. the font size of the subsection caption is bigger than the one of the (main-)section.

**It is standard Copernicus latex template size for subsections written in latex as:**

**\subsection{Dry/wet classification and baseline estimation}**

-- Section 4 – In my opinion it is a bit too extensive, since it is mostly hypothetical. It is more extensive than the discussion of the data quality and the variable description. This is not the right relation for a data paper. In this way it kind of degrades the more important sections, which are on the other hand a bit short.

**This point is related to one of the comments from the first referee. Based on that, we decided to restructure chapter 4 into two subsections (former application and outlook). We consider chapter 4 as an important part to present the potential of the dataset. Nevertheless, the discussion of the data quality was extended in the running text and by new metadata sheets as mentioned before.**

Final remarks:

The comments should not hide the fact that the paper is a good data publication. The data are of high scientific interest and the data availability and quality appear to be very good. The paper has also been written in an appealingly clear manner. However, some important points should be changed or expanded in order to provide the best possible basis for future users of the data. What is meant above all is a significant expansion of the metadata and a more detailed discussion of the data quality.

---

## Author Response (AR2)

**Response to the technical corrections**

We thank both referees for reviewing our article. We answer technical corrections of the referee #1 below. Our answers are in **bold**.

I have had the opportunity to review the first draft of this manuscript and I appreciate the changes and improvements the authors have made in this second draft. I do not see major issues in this version but rather some minor, mostly textual changes which I listed below.

Comments (line numbers refer to the tracked changes manuscript)

l. 8-10: The two sentences are repetitive regarding the data quality, I suggest to delete the second one.
**The second sentence was deleted.**

l. 15: To give examples of the potential of CML rainfall estimates the country-wide analysis for Germany and the Netherlands could be referenced here. (Graf et al., 2020; Overeem et al., 2016)
**References were added.**

l. 17: Rephrase to: "CMLs can observe..."
**Rephrased.**

l. 43-45: The data quality section should also be mentioned here
**The data quality section was mentioned in the text.**

**The changes are in red:**
**"…experimental campaign,  structure of the collected datafiles, and discuss data quality and reliability. The third…"**

l. 49: rephrase to: "The attenuation of a CML signal is related to the drop size distribution along its path. The observed attenuation can be used to calculate the rain rate between the two end nodes of a CML."
**Rephrased.**

l. 68: rephrase to "volume in a drop diameter interval"
**Rephrased.**

l. 78: please use "CML" or "microwave link" throughout the manuscript
**"CML" replaced "microwave link" here and also in several other parts in the text.**

l. 84: rephrase sentence
**The changes are in red:**
**"Three tipping bucket rain gauges were also used at sites 2, 4 and 5 (Table 1)."**

l. 101: the abbreviation "EM" is undefined
**Abbreviation explained when used for the first time in text.**

l. 104: rephrase to: "The CML path was not horizontal"
**Rephrased.**

l.116: rephrase the second part of the sentence
**The changes are in red:**
**"The cob webs were not specifically prevented and they were not observed during the field visits ."**

Caption Figure 2: MiniLink is written in a different way than on l. 97
**"MINI-LINK" is now used consistently in the manuscript.**

l. 146: rephrase to "sheets could unfortunately not be compiled from"
**Rephrased.**

l. 167: add a comma after "sunlight duration"
**Added.**

l. 167: replace time step with resolution as this word is used before
**Replaced.**

Section 2.5: refer to Appendix J in this section
**Following sentence was added to the end of this section:**
**"The Appendix J presents the data availability of the devices during the campaign period."**

l. 246: replace "built environments" with "surrounding buildings"
**Replaced.**

l. 255: rephrase the sentence "If we wanted to get .." as it is too colloquial
**The changes are in red:**
**"Heated antennas would be needed to prevent the formation of dew and/or air blowers  would blow away the water during fog events."**

l. 257: state why it remains a challenge
**The changes are in red:**
**However, modelling these effects remains challenging, since the dew-related wetting depends on the weather conditions, as well as on the condition/presence of hydrophobic radome cover.**

l. 274: delete "the"
**Deleted.**

l. 302: replace "inputting" with "interpolating"
**Replaced.**

l. 341: delete the redundant "and"
**Deleted.**

Graf, M., Chwala, C., Polz, J., & Kunstmann, H. (2020). Rainfall estimation from a German-wide commercial microwave link network: Optimized processing and validation for 1 year of data. Hydrology and Earth System Sciences, 24(6), 2931–2950. https://doi.org/10.5194/hess-24-2931-2020
Overeem, A., Leijnse, H., & Uijlenhoet, R. (2016). Two and a half years of country-wide rainfall maps using radio links from commercial cellular telecommunication networks. Water Resources Research, 52(10), 8039–8065. https://doi.org/10.1002/2016WR019412